# Fast and stable color normalization of whole slide histopathology images using deep texture and color moment matching

Kyohei Sano[1], Daisuke Komura[1], Shumpei Ishikawa[1]

[1] Department of Preventive Medicine, Graduate School of Medicine, The University of Tokyo, 7-3-1 Hongo, Bunkyo-ku, 1130033, Tokyo, Japan.

**Abstract.** Whole Slide Images (WSIs) are prone to color variations due to differences in fixation and staining conditions of tissue samples, as well as the scanning process. Such variations can adversely affect the image analysis, and in this paper, we propose a novel, fast and stable color normalization algorithm for WSIs called CONTEMM (COlor Normalization using deep TExture and color Moment Matching). CONTEMM estimates color transformation matrix based on pairs of reference and source patches with similar tissue components in the respective WSIs, which are selected using deep texture representations. The color transformation matrix is estimated quickly by fitting the second moment about white color.

Performance of CONTEMM algorithm was evaluated using histopathology images from different slide scanners and TCGA (The Cancer Genome Atlas) datasets. CONTEMM was shown to outperform the other methods; Reinhard, Vahadane, and Macenko, in terms of variation (stability), accuracy, and computation time.

## 1      Introduction

In histopathology, tissue sections are stained with multiple contrasting dyes (e.g., the most widely used hematoxylin and eosin (HE) stain) to highlight different tissue structure and cellular features, and pathologists make diagnosis of diseases under the microscope. However, histopathological images contain undesired variability of HE stain appearance due to differences in fixation, staining procedures, and scanners. Color normalization methods, which reduce the color variation of source images using a reference image, are often effective to improve the performance of histopathological image analysis.

Although various color normalization methods have been developed so far, most methods focus on normalization of patches sampled from WSIs and there are few methods optimized to gigapixel-sized whole slide images (WSIs). For example, Reinhard *et al* [1] proposed a patch normalization method, which is aimed at matching the color distribution of source patches to a reference patch in L*a*b* color space. This algorithm is quite fast, but assumes that the source and reference patches are composed of similar tissue, which does not hold generally in WSIs. When the method is applied to

all patches of WSIs, the performance lacks stability due to the different tissue compositions between source and reference patches.

Macenko *et al*[2] and Vahadane *et al*[3] proposed color normalization methods based on stain deconvolution, which estimates hematoxylin and eosin vectors from the distribution of color space. This estimation is more robust against the tissue composition difference between source and reference patches, but it requires longer computation time than Reinhard *et al*. Thus, analyzing thousands of WSIs, which is common recently, is almost infeasible without high computational power. Also, these color normalization methods fail when tissue compositions of the reference and the source patch differ significantly. In addition, Macenko's method is also a patch-based normalization method and suffers from the same problem as the Reinhard's method.

To tackle these problems, we propose a novel color normalization algorithm for WSIs called CONTEMM (COlor Normalization using deep TExture and color Moment Matching), which is significantly stable and fast. Instead of using the whole images, CONTEMM selects appropriate pairs of image patches with similar tissue components in reference and source WSIs based on deep texture representations (DTRs). CONTEMM estimates global transformation matrix between the pairs and normalization of the source WSI is performed using the transformation matrix. One notable feature of CONTEMM is that it achieves high-speed color normalization by a simple linear transformation to fit the second moment about white color, which match the stain vector without doing stain deconvolution.

## 2    Method

### 2.1    Overview of the CONTEMM

CONTEMM searches the most similar regions from source and reference WSIs, and calculate the transformation matrix between them. This transformation matrix is then applied to the source images for color normalization. Figure 1 shows the overview of CONTEMM, which is consisting of the following three steps.

- STEP I: Operation in reference WSI

  a. N patches are randomly sampled from a reference WSI.
  b. Deep texture representations are extracted from all N patches using deep convolutional network. (2.1)

- STEP II: Operation in source WSI

  a. n patches are randomly sampled from a source WSI.
  b. Deep texture representations are extracted from all n patches using deep convolutional network. (2.1)
  c. Pair up N and n patches based on texture features to form predetermined number (m) of the most similar-looking pairs.
  d. A transformation matrix is calculated using the pairs. (2.2)

- STEP III: Operation in source images for color normalization.

a. The transformation matrix is applied to source images. (2.2)

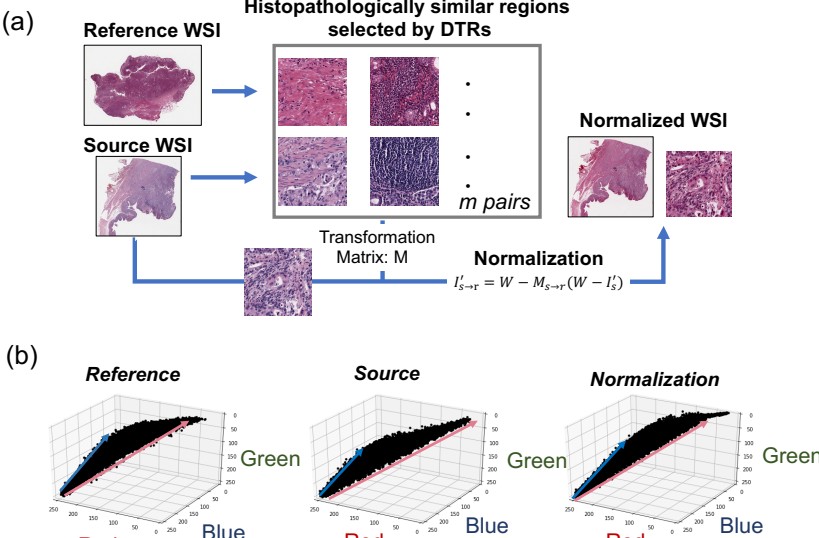

**Fig. 1.** Overview of the CONTEMM color normalization method. (a) Schematic of CONTEMM. (b)Transformation in RGB space

Quantitative metrics representing image similarity is necessary to search for the most similar regions in source and reference WSIs. We use second order statistics within deep features called deep texture representations (DTRs). Deep texture representations extracted from pre-trained convolutional neural network (CNN) are often used to calculate the perceptual similarity in general images due to the robustness to image distortion. The DTRs further produces order-less image representations suitable for searching similar histopathology images as shown previously[4]. Here, the output of $9^{th}$ convolution layer ("block4_conv2") in VGG16 [5], which is often used for perceptual similarity in general images, is used to compute the Gram matrix $G^l \in \mathcal{R}^{N_l \times N_l}$ by bilinear pooling using the following equation:

$$G_{ij}^l = \sum_k F_{ik}^l F_{jk}^l .$$

Here $F_{ik}^l$ is the vectorized activation of the $i^{th}$ filter at position $k$ in in layer $l$. In order to reduce the time required to search for the most similar regions, their dimensions are reduced by Compact bilinear pooling (CBP)[6] from 262144 dimensions to 1024 dimensions, and cosine similarity of the CBP output was used as a similarity measure as in [4].

4

## 2.2     Transformation Matrix

In CONTEMMN, RGB vector in source image is rotated and scaled at the center of white color to fit the second moment about white to the reference patches (Fig.1b). Let $I_r, I_s \in R^{h \times l}$ be the matrix of RGB intensities of reference and source patches chosen by similarity search respectively, where h = 3 for RGB channels, and l = total number of pixels of m patches, and let W $\in R^{h \times l}$ be the matrix of 255s, which represent white color in 8 bit RGB color. Let $X_r, X_s \in R^{h \times l}$ be the matrix of RGB intensities with the origin located at the coordinate of white color. Then $X_r$ and $X_s$ can be written as follows,

$$X_r = W - I_r, X_s = W - I_s$$

Let $C_r, C_s \in R^{h \times h}$ be the matrix of the second moment about white color of target and source patches respectively.

$$C_r = X_r X_r^T, \qquad C_s = X_s X_s^T$$

and the eigenvalue decomposition of $C_r$ and $C_s$ are given as

$$C_r = P_r \Lambda_r P_r^{-1}, C_s = P_s \Lambda_s P_s^{-1}$$

Let $\Theta_r$ and $\Theta_s$ be diagonal matrix, and $\Lambda_r$ and $\Lambda_s$ can be factorized as follows,

$$\Lambda_r = \Theta_r \Theta_r, \qquad \Lambda_s = \Theta_s \Theta_s$$

Transformation matrix $M_{s \to r}$ can be written,

$$M_{s \to r} = P_r \Theta_r \Theta_s^{-1} P_s^{-1}$$

Let $I'_s, I'_{s \to r} \in R^{h \times k}$ be the matrix of RGB intensities of a source image for color normalization and the image after color normalization, where k = number of pixels of a source image then,

$$I'_{s \to r} = W - M_{s \to r}(W - I'_s)$$

Fitting the second moment about mean, variance and covariance, is a major color transfer method[1][7]. However, these methods don't work well in white color transfer. In addition, color deconvolution is one of the most widely used methods for stain normalization[3], and fitting the second moment about white can match the stain vector of reference and source images (Fig.1b).

# 3 Results and Discussion

## 3.1 Hyperparameter selection

Generally, there is one reference WSI and multiple (possibly thousands of) source WSIs for each task. Since step 1 is performed only once, N can be large without influencing the total computation time much. In contrary, n cannot be large because step 2 is performed for every source slide. m/n should be appropriate, because it can increase the chance of selecting artifact regions as similar pairs, especially when WSIs contain a lot of artifact. The size of patches should not be large because it takes long time to read large patches. Thus, we have set N=1000, n=40, m=30 in Experiment 1 and 3, and N=1000, n=40, m=15 in Experiment 2, and all patch size was set to $256 \times 256$ pixels. Here we decrease m value in Experiment 2, because some WSIs have a lot of artifacts such as pen marks.

## 3.2 Experiment 1: Quantitative evaluation (PSNR)

First, accuracy and stability of CONTEMM was evaluated. Each of three WSIs of stomach adenocarcinoma was scanned using two different slide scanners (Hamamatsu photonics NanoZoomer S60 (Hamamatsu) and 3D HISTECH Pannoramic MIDI II (3DX)) and the performance was evaluated in a pixel-wise manner.

CONTEMM was compared with the other three normalization methods; Reinhard [1], Macenko [2], and Vahadane [3]. In Vahadane et al, there are two color normalization methods proposed, which we call "Vahadane (random)." and "Vahadane (WSI)". In Vahadane (random), source patches is normalized to one specific reference patch. In Vahadane (WSI), a WSI was split by grid and reference patches were sampled at grid points. In this experiment, a WSI was divided into 5x5 grids.

In, Reinhard, Macenko, and Vahadane (random), the reference patch was randomly sampled to each source patch, excluding the white background patches. Patches with the median RGB value greater than 220 were regarded as white background.

Mean and variance of Peak Signal-to-Noise Ratio (PSNR) improvement were used to assess the accuracy and the stability of the color normalization method, respectively. 70 patches were randomly sampled from the same position in source and reference WSIs, and the same 70 patches are used for evaluation in all color normalization methods. Since there was unignorable difference in sharpness between images from two scanners, Gaussian filter was applied before color normalization in all methods to reduce the effect of sharpness on PSNR. The size of the Gaussian filter was optimized to match the sharpness, which was estimated by Laplacian Kernel[8].

Registration between two WSIs is performed by imreg_dft package [9]. As each three WSIs from a scanner was color-normalized to the other one and vice versa, six transformations were obtained in total.

As shown in Table 1, CONTEMM significantly outperforms Macenko, Vahadane (random), and Reinhard in terms of both accuracy and stability. CONTEMM also significantly outperforms Vahadane (WSI) in terms of stability with comparable accuracy.

We also investigated the failure patterns in this experiment. In Figure 2, and Figure 3, two types of failure were observed: first, Reinhard, Macenko, Vahadane (random) and Vahadane (WSI) failed when the background of source image was white (Fig. 2a, Fig.3). Second, Reinhard, Macenko and Vahadane (random) did not work when the appearance of reference patch is significantly different from the source patch (Fig. 2b).

**Table 1.** The improvement of PSNR: Mean ± Standard Deviation. **H**: Hamamatsu photonics NanoZoomer S60. **3DX**: 3D HISTECH Pannoramic MIDI II. The number in a square bracket corresponds to the slide number. Dunnet contrasts tests and F-tests were used to compare the mean and the variance of PSNRs of CONTEMM and those of the other methods, respectively. Bonferroni correction was applied for F-tests.

| Source-> Reference | CONTEMM (Ours) | Vahadane (WSI) | Macenko | Vahadane (random) | Reinhard |
|---|---|---|---|---|---|
| 3DX->H[1] | -0.082±**0.718** | **0.682**±1.043 | -0.266±1.784 | -0.984±2.187 | -0.964±2.462 |
| 3DX->H[2] | 2.379±**1.302** | 3.235±1.511 | **3.259**±3.297 | 1.424±3.532 | -0.491±4.905 |
| 3DX->H[3] | 2.293±**0.808** | -0.586±1.597 | **2.695**±3.053 | 1.214±2.884 | 1.111±3.160 |
| H->3DX[1] | -0.199±**1.302** | **0.495**±1.327 | -1.918±2.249 | -2.092±2.458 | -2.741±2.399 |
| H->3DX[2] | 0.445±**0.927** | **3.597**±1.948 | -1.378±2.590 | -1.593±3.397 | -2.930±3.967 |
| H->3DX[3] | **0.340**±**0.756** | -4.425±1.454 | -0.697±1.457 | -0.010±2.499 | -0.587±2.321 |
| Mean | **0.863**±**1.461** | 0.499±3.059 | 0.282±3.183 | -0.340±3.165 | -1.100±3.619 |
| Pvalue (mean) | N/A | $p > 0.05$ | $p < 0.05$ | $p < 0.001$ | $p < 0.001$ |
| Pvalue (variance) | N/A | $p < 0.001$ | $p < 0.001$ | $p < 0.001$ | $p < 0.001$ |

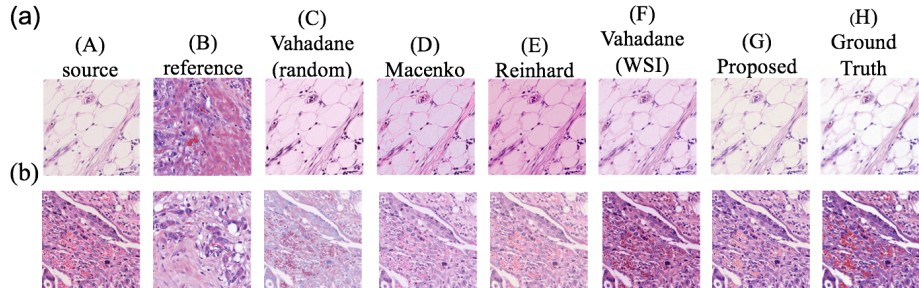

**Fig. 2.** Examples of the failure. (a) Image with large white regions. (b) Source and reference patches quite different from each other. (A): source patch, (B): reference patch for (C)-(E), (C): Vahadane(WSI), (D): Macenko, (E): Reinhard, (F): Vahadane(random), (G): CONTEMM, (H): Ground Truth. (C)-(E): Source patch was color-normalized using reference patch (B). Note that (B) is not used as reference patch in (F), (G).

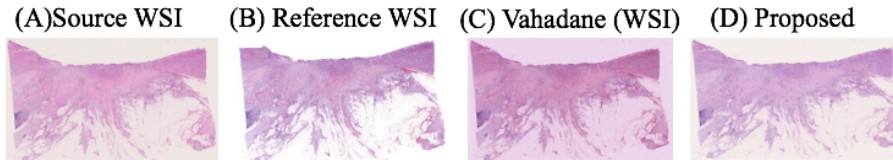

**Fig. 3.** The appearance of color normalization of thumbnail. Source WSI is scanned by Hamamatsu scanner and Reference WSI is scanned by 3D HISTECH.

### 3.3 Experiment 2: Quantitative evaluation (NMI)

Next, we evaluated the consistency of normalization using 100 randomly selected WSIs of kidney renal clear cell carcinomas from The Cancer Genome Atlas (TCGA) datasets[10]. WSI-level normalization was performed using CONTEMM and Vahadane (WSI) and the normalized median intensity (NMI) within uniform tumor region selected by pathologists were compared. Lower standard deviation of NMI (NMI SD) indicates that the normalization is more consistent [11]. As shown in Table 2, CONTEMM showed better NMI than Vahadane (WSI) and the original WSIs.

**Table 2.** Standard deviation of the normalized median intensity.

|  | CONTEMM | Vahadane (WSI) | Original WSIs |
|---|---|---|---|
| NMI SD | **0.0677** | 0.0685 | 0.0749 |

### 3.4 Experiment 3: Computation time evaluation

Finally, computation time of each color normalization step was measured using a source slide. One Tesla V100 GPUs and Dual Intel 20-core Xeon E-2698v4 2.20GHz was used for computation. Table 3 shows that our algorithm is significantly fast. It takes only 96 seconds for step 1, 3.16 seconds for step 2, and 0.0034 seconds for step 3. Fast color normalization method is especially important recently, as the number of WSIs being analyzed has grown dramatically. For example, more than 10,000 WSIs are analyzed in a recent study[12]: it would take around 1,500 hours to standardize 10,000 WSIs using Vahadane (WSI), while it would take only about 9 hours using CONTEMM, which is feasibly short.

**Table 3.** Computation time of color normalization: "WSI (sec)" is the computation time required for estimating the color transformation between two WSIs. "patch (sec)" is the computation time required for color-normalizing one patch. Macenko, Vahadane (random), and Reinhard do not have WSI-level color normalization.

|  | CONTEMM | Vahadane (WSI) | Reinhard | Macenko | Vahadane (random) |
|---|---|---|---|---|---|
| **WSI (sec)** | **3.16** | 537.9 | N/A | N/A | N/A |
| **patch (sec)** | **0.0034** | 0.013 | 0.0075 | 7.04 | 8.05 |

## 4 Conclusion

In this paper, we proposed CONTEMM, a fast and stable color normalization method for WSIs in histopathology. CONTEMM estimates a global transformation matrix

based on reference and source patch pairs selected based on deep texture representations. Experimental results showed that CONTEMM outperforms the existing patch-based methods in terms of stability and accuracy. Additionally, CONTEMM outperforms a WSI-based method, Vahadane (WSI), in terms of stability and computation time while keeping comparable accuracy. Notably, compared to the WSI-based method, CONTEMM speeds up the color normalization process of WSIs by several orders of magnitude, which makes it feasible to normalize thousands of WSIs in realistic time. CONTEMM would be a powerful tool for histopathology image analysis in the big data era, where rapid color normalization is essential.

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
