# OpenReview forum: "Fast and stable color normalization of whole slide histopathology images using deep texture and color moment matching"
_MICCAI.org/2019/Workshop/COMPAY — COMPAY 2019_

### Official Review · AnonReviewer3 · 2019-08-11
**Simple yet good idea, but needs some more validation**

**Rating:** 6
**Confidence:** 3

**Review:**

Summary:
The authors present a new method for color normalisation of histology images where image patches from a source WSI are normalised to image patches from a reference WSI based on similarity in texture.

Detailed comments:
This is a simple idea, yet seems to be beneficial in terms of the speed of normalisation. It is true, that alternate strategies can be very time consuming and serve as a significant computational overhead. I think the method has potential, but I would recommend addressing the following comments:

It is clear that patch based normalisation techniques are sensitive to the choice of target patch. This is addressed your paper by the selection of similar image patches based on texture similarity. In my personal experience, I have seen that the Vahadane normalisation algorithm does a decent job and therefore question the choice of target image patch selected to obtain the results. A random image patch would not typically be selected, but an appropriate image patch that leads to good performance. My main concern is that a bad image patch may have been selected to highlight positive results. I think for an appropriate head to head comparison, it is imperative to add the experiment of Vahadane & Macenko but with the same target image patches used via texture similarity. If similar results are then obtained using CONTEMM, but the speed is significantly better then this will be a positive contribution of the work.

Please add a line to explain the task of the VGG network that was used to assess texture similarity.

Finally, if you intend this paper to be more impactful, then it will be necessary to add some more up-to-date comparison with stain normalisation methods (or at least mention them in the text):
- Khan, N. Rajpoot, D. Treanor, and D. Magee, “A nonlinear mapping approach to stain normalization in digital histopathology images using image-specific color deconvolution,”
- B.E. Bejnordi, G. Litjens, N. Timofeeva, I. Otte-Hller, A. Homeyer, N. Karssemeijer, and J. AWM van der Laak, “Stain specific standardization of whole-slide histopathological images,” IEEE transactions on medical imaging, vol. 35, no. 2, pp. 404-415, 2016.
- Zanjani, Farhad Ghazvinian, et al. "Stain normalization of histopathology images using generative adversarial networks." 2018 IEEE 15th International Symposium on Biomedical Imaging (ISBI 2018). IEEE, 2018.

---

### Official Review · AnonReviewer2 · 2019-08-14
**No title**

**Rating:** 6
**Confidence:** 3

**Review:**

This paper presents a stain normalization method based on deep convolutional features and color transformation matrix. The proposed algorithm referred as CONTEMM and contains 3 main steps, a) compute deep texture representations of randomly sampled patches from reference and source whole slide images (WSIs), (b) select patches based on their cosine similarity to compute transformation matrix, and (c ) apply transformation matrix for color normalization. This study is performed on 3 stomach adenocarcinoma slides scanned with two different scanners and 100 randomly sampled WSIs of kidney renal cell carcinomas from The Cancer Genome Atlas (TCGA) dataset. Overall, the proposed method has outperformed some of the previously published methods and requires less time to process a WSI.

Below are some comments and suggestions

- The reported accuracy and stability of CONTEMM are based on 3 WSIs so it is important to discuss how the performance of this method would be scaled on larger datasets. Limitations or challenges (if any) should be discussed
- In section 2.1, the term ‘block4_conv2’ is not very descriptive and need further clarifications
- What type of matrix is G? The variables involved in equation 2.1 needs to clear explanation.
- In section 3.4, it is not clear that reported computation time is based on one specific whole-slide image or is it the average of multiple images
- DOI for VGG-Net (5th reference) is incorrect
- It is not clear which package is imreg_dft, a proper citation would be needed
- In section 1, the second paragraph it should be L*a*b* instead of 𝑙𝛼𝛽
- If possible, use ‘equation editor’ for distinguishing text from variables and symbols

---

### Official Review · AnonReviewer4 · 2019-08-15
**Method needs comparison to more recent approaches.**

**Rating:** 5
**Confidence:** 3

**Review:**

Summary:
A method for color normalization of whole slide images. The method pairs image from source and reference based on similarity of VGG16 texture features. A transformation matrix is then calculated based on the paired images to be applied to the source images. The authors compare their method with some other stain normalization methods.

The proposed method of creating a transformation matrix is similar to previous approaches, but has a faster computation time, which can be beneficial when little resources are available. The sampling strategy is an interesting method for choosing reference patches.

I have concerns about the comparison with the other methods (Vahadane, Reinhard, Macenko). The author compares with other method's using different sampling strategies. It would be quite interesting to see the effect of using the sampling strategy based on the VGG16 texture features. This would allow for discernment between the effect of the author's normalization matrix and sampling strategy on the performance.

Finally, I would suggest the authors to compare with at least one of the more recent stain normalization methods that have also shown to be better than Vahadane, Reinhard & Macenko's methods. The methods that the authors compare to, however well-established and useful, would not be considered state-of-the-art anymore. Some suggestions:
1. Bug, Daniel, et al. "Context-based normalization of histological stains using deep convolutional features." Deep Learning in Medical Image Analysis and Multimodal Learning for Clinical Decision Support. Springer, Cham, 2017. 135-142.
2. Shaban, M. Tarek, et al. "Staingan: Stain style transfer for digital histological images." 2019 IEEE 16th International Symposium on Biomedical Imaging (ISBI 2019). IEEE, 2019.
3. Zanjani, Farhad Ghazvinian, et al. "Stain normalization of histopathology images using generative adversarial networks." 2018 IEEE 15th International Symposium on Biomedical Imaging (ISBI 2018). IEEE, 2018.
4. Bejnordi, Babak Ehteshami, et al. "Stain specific standardization of whole-slide histopathological images." IEEE transactions on medical imaging 35.2 (2015): 404-415.

Specific comments:
- Section 1 "A transformation matrix is calculated using the pairs. (2.3)", there is no 2.3, or did you mean 2.2 here?
- Section 2.1 "CONTEMM was compared with three state-of-the-art color normalization methods..", the methods compared to in the paper are not state-of-the-art anymore.
- Section 3.2 "As three WSIs from a scanner was color-normalized to the other one and vice versa,
six transformations were obtained in total.", that would result in a total of nine conversions. From table 1 it seems you only picked one slide from the 3D HISTECH scanner. Was there any reason for this?
- Section 3.2 "Gaussian filter was applied to adjust the sharpness to reduce the effect on PSNR", was this gaussian filter applied after or before normalization? Was it applied for all methods?
- Section 3.3 70 patches for evaluation were randomly sampled from the same position in source and reference WSIs. Were the same 70 patches evaluated for each method? Please clarify.

Conclusion:
Overall, the method needs comparison to more recent approaches. The sampling strategy seems like an interesting approach for sampling patches from paired whole slide images. As it stands, it is unclear how the different parts of the author's method (sampling strategy & normalization matrix) affect the performance.

---

### Decision · Program_Chairs · 2019-08-20

Accept